# MUIQD: Benchmarking and Facilitating Multimodal LLMs for Underwater Image Quality Perception

## Abstract

Multimodal Large Language Models (MLLMs) have recently demonstrated great potential in cross-modal perception, reasoning and generation. However, their effectiveness in underwater image quality perception, a fundamental requirement for efficient underwater vision tasks, remains largely unexplored. To address this gap, in this paper we propose the first large-scale dataset for benchmarking and facilitating underwater image quality perception of MLLMs, termed MUIQD. MUIQD is composed of two complementary subsets, MUIQD-Description and MUIQD-VQA, addressing the abilities of quality perception and interaction of MLLMs, respectively. Specifically, MUIQD-Description comprises 18634 underwater images covering diverse real-world underwater scenes and typical quality degradations, such as color cast, haze effect, blurring and low contrast, etc. Each image is annotated through rigorous subjective evaluation with detailed descriptions of quality-related attributes and an overall quality level derived from the attributes. To further improve the interactive quality perception capability of MLLMs, we build the visual-question-answering-based dataset MUIQD-VQA, containing more than 93K question-answer pairs derived from the MUIQD-Description dataset generated by DeepSeek. Experimental results demonstrate that the proposed MUIQD dataset promotes the abilities of MLLMs on underwater image quality perception significantly, strongly supporting that MLLMs can be adapted for underwater image quality perception.

## 1 Introduction

Underwater imaging plays a vital role in a variety of underwater tasks, such as resource exploration, marine monitoring and biological conservation, etc. In comparison with imaging in the atmosphere, underwater imaging encounters some great challenges, e.g., light absorption and scattering, insufficient lighting, water fluctuation, which make the obtained image susceptible to color cast, fog, blur, low contrast, degrading the image quality severely. Therefore, accurate perception of the underwater image quality becomes a critical prerequisite for the underwater imaging-based tasks.

In recent years, with rapid developments of deep learning technologies, Multimodal Large Language Models (MLLMs) that integrate information from multiple modalities (e.g., text, image, audio, video, etc.) have demonstrated strong cross-modal comprehension and reasoning abilities, highly aligning with the way that humans perceive and comprehend the external information. However, the MLLMs' perception ability for the underwater image quality still remains unexplored. More specifically, whether MLLMs possess such an ability or whether this ability can be further perfected are both still unclear.

With the above queries in mind, in this paper we make the first attempt to construct a large-scale MLLM-oriented underwater image quality dataset (MUIQD) for fundamentally benchmarking and facilitating MLLMs for underwater image quality perception. The constructed MUIQD is made up of two subsets, which are MUIQD-Description and MUIQD-VQA, addressing the MLLM's abilities of quality description and interactive quality perception for underwater images. For constructing the MUIQD-Description subset, we collect a total of 18634 authentic underwater images and carry out extensive subjective tests to provide direct quality feedback in terms of text for each image. The

viewers' feedback contains two parts. The first part refers to a detailed description of the essential quality attributes (e.g. color cast, contrast, blur and noise, exposure, etc.) and the details or contexts (e.g. unclear details of the fish) that closely relate with the image quality. The other part refers to an overall quality evaluation conclusion deduced from the descriptions.

In order to make MLLMs adaptable to various quality queries, we further convert the detailed descriptions in MUIQD-Description to question-answer pairs with DeepSeek-V3 Liu et al. (2024a), producing a specific visual-question-answering-based dataset MUIQD-VQA. In MUIQD-VQA, each image is associated with five questions about its quality and four quality-related attributes, i.e., color cast, contrast, blur and noise, and exposure. Each attribute-question type is randomly selected from three types, i.e., Yes-or-No, What, and How, as adopted in Wu et al. (2024). Through this manner, we generate more than 93K question-answer pairs which are used to examine and fine-tune the MLLMs for interactive quality perception. Compared with the existing traditional underwater image quality datasets Liu et al. (2024e)Yang et al. (2021)Hou et al. (2023), MUIQD-description contains a total of 18634 underwater images, which cover abundant underwater scenes and quality-degradation situations. In addition, we annotated the underwater image essential attributes, details and the overall quality level, rather than only a single number that indicates the image quality adopted by existing underwater image quality databases. What's more, we generate more than 93K question-answer instructions for tuning MLLMs to perceive the underwater image quality more effectively. Therefore, the proposed MUIQD is the first MLLM-oriented underwater image dataset for benchmarking and facilitating MLLMs for underwater image quality perception. We summarize the contributions of this paper as follows:

- We construct the first large-scale MUIQD-Description dataset, which includes 18634 underwater images with detailed descriptions of the image essential attributes and overall quality level, specially benchmarking and facilitating MLLMs for underwater image quality perception.

- Based on MUIQD-Description, we construct MUIQD-VQA, which is the first visual-question-answering dataset for tuning MLLMs towards dealing with various underwater image quality queries.

- We fine-tune some SOTA MLLMs on the proposed MUIQD and verify that MUIQD can boost the fundamental abilities of MLLMs for underwater image quality perception significantly, which fully confirms that MLLMs can be adapted for underwater image quality perception.

## 2 RELATED WORKS

### 2.1 UNDERWATER IMAGE QUALITY PERCEPTION MODELS AND DATASETS

#### 2.1.1 QUALITY PERCEPTION MODELS

Existing underwater image quality perception methods concentrate on predicting an image quality score that indicates the image quality level through computational models. For example, Yang et al. proposed the UCIQE metric which quantifies the colorfulness, contrast and saturation and fused them linearly to reach an overall quality score Yang & Sowmya (2015). Panetta et al. established an underwater image quality perception model by characterizing the image colorfulness, sharpness and contrast Panetta et al. (2016). Wang et al. constructed the CCF metric which computes the image quality score through measurements of colorfulness, contrast and fog density Wang et al. (2018). Yang et al. designed quality-aware features from the image frequency domain to infer the underwater image quality level Yang et al. (2021). Liu et al. proposed a more comprehensive quality evaluation index by characterizing more quality-related attributes Liu et al. (2024d). Zheng et al. constructed the UIF metric by quantifying the image naturalness, clarity and structural loss Zheng et al. (2022). Based on the deep learning technologies, Liu et al. designed an end-to-end underwater image quality predictor which integrates attention mechanisms and Vision Transformer (ViT) Dosovitskiy et al. (2020) to characterize the channel, spatial and global appearance of the image for quality prediction Liu et al. (2024e). Wang et al. learned the luminance and chrominance feature representations jointly based on generation strategy to estimate the image quality Wang et al. (2022). URanker Guo et al. (2023) exploited histogram prior and cross-scale correspondence

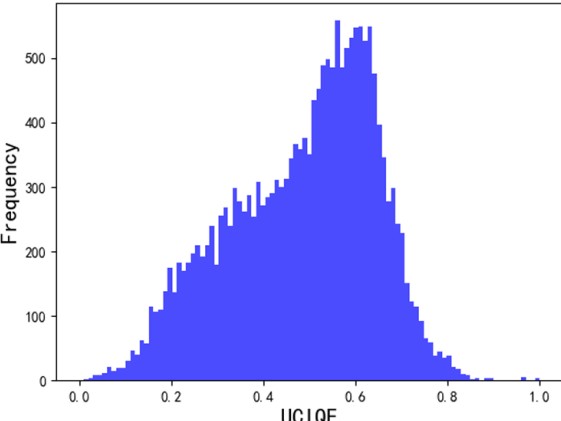

Figure 1: Quality distribution of the underwater images measured by the UCIQE metric Yang & Sowmya (2015).

| Dataset | Subset / Image Num. | Sampled Image Num. |
|---|---|---|
| UIISLian et al. (2023) | UDW / 4628 | 4628 |
| UIQDLiu et al. (2024e) | UIQD / 5369 | 5369 |
| UIEBDLi et al. (2019) | Raw / 890 | 884 |
| SUIMIslam et al. (2020a) | train_val / 1525 | 1517 |
| EUVPIslam et al. (2020b) | Unpaired / 6665 | 6236 |

Table 1: Information of the image datasets for sampling.

to capture global and local degradation of the underwater image. Wang et al. designed a prior-based underwater image quality scoring method, which integrates relevant imaging information into the quality prediction framework Wang et al. (2024). With rapid advancements of vision-language models (VLM), VLM-based image quality perception methods have been investigated Wu et al. (2025)Wu et al. (2023b). In Zhang et al. (2023), Zhang et al. employed the VLM to learn the visual-language correspondence and implemented image quality perception via carefully-designed multitask learning strategy. In Wu et al. (2023b), Wu et al. calibrated the LMMs to perceive the image quality via text-defined rating levels. In You et al. (2025), You et al. used score discretization to calibrate LLMs for more accurate quality perception. In Li et al. (2025), Li et al. proposed a reinforcement learning-based framework based on group relative policy optimization and improved the quality reasoning ability of the MLLMs. In Wu et al. (2025), Wu established a reasoning-induced image quality perception model VisualQuality-R1, which was calibrated with reinforcement learning to rank and demonstrated superior performance of quality reasoning and prediction.

### 2.1.2 DATASETS

To facilitate underwater image quality perception, some representative databases have been built. Wang et al. captured 87 underwater images by photographing a standard color card in water to generate an underwater image quality database Wang et al. (2018). Li et al. proposed an underwater image database of 950 underwater images for facilitating underwater image quality perception and enhancement Li et al. (2019). Yang et al. built a specific database of 890 underwater images to test the prediction performance of underwater image quality evaluation methods Yang et al. (2021). Liu et al. set up a multi-view underwater image acquisition system and formed a large-scale underwater image database to examine different aspects of enhancement algorithms, e.g., quality assessment, color cast correction and high-level machine vision capabilities Liu et al. (2020). Liu et al. proposed a relatively large-scale database, which contains 5369 authentic underwater images covering rich underwater scenes and typical distortion types Liu et al. (2024e). Guo et al. reported an underwater image database for enhancement Guo et al. (2022), which was formulated by enhancing 40 underwater images with five enhancement methods. Wang et al. enhanced 2000 underwater images by

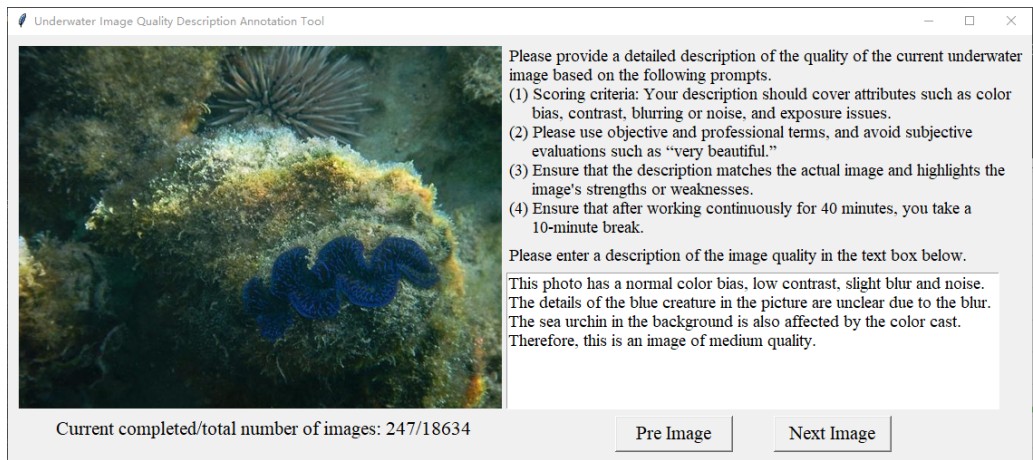

Figure 2: Subjective experiment interface.

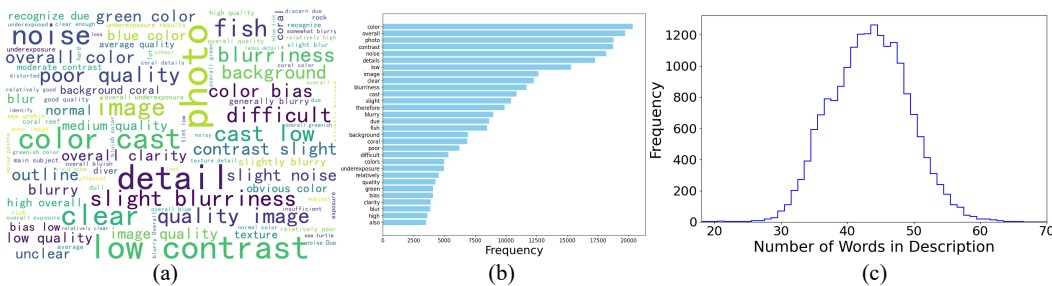

Figure 3: Statistics of the subjective quality descriptions. (a) word cloud; (b) frequency of the words; (c) distribution of the description length.

15 enhancement schemes, leading to an image database of 30000 images Wang et al. (2022). Hou et al. constructed an underwater image database Hou et al. (2023) by enhancing 60 underwater images with 15 enhancement models. Jiang et al. established a database of 1000 enhanced underwater images for enhancement Jiang et al. (2022). Zheng et al. chose 100 typical underwater images and enhanced them with 10 enhancement methods Zheng et al. (2022). Guo et al. generated 8010 enhanced underwater images and adopted the paired-comparison methodology to annotate the image quality Guo et al. (2023). The above underwater image databases adopt traditional methods to annotate the image quality with a single quality score, i.e., mean opinion score (MOS), which lacks both detailed description of the image quality and the important quality reasoning process, resulting in inadequacy to calibrate the MLLMs to perceive the underwater image quality Wu et al. (2024).

## 2.2 MULTIMODAL LARGE LANGUAGE MODELS

MLLMs are capable of processing and comprehending information from multiple modalities, offering powerful technical support for cross-domain applications. In the early age, researchers focused on fusing information from different modalities into a unified framework. For instance, Chen et al. constructed an image description dataset to explore combining image information with natural language Chen et al. (2015). CLIP Radford et al. (2021) and ALIGN Jia et al. (2021), as classical multimodal large models, realized the joint representation of vision and text through contrastive learning. DALL-E Reddy et al. (2021) and Stable Diffusion Rombach et al. (2022) addressed the task of generating high-quality images from text, demonstrating great potential of MLLM on generative tasks. With the advent of GPT-4 OpenAI (2023), LLMs have ushered in a period of rapid development, based on which researchers have further explored the capabilities of MLLMs. Currently, MLLMs Bai et al. (2025); Lu et al. (2024); Chen et al. (2024); Hong et al. (2024); Liu et al. (2024b); Team et al. (2025) have been successfully employed to handle various multimodal tasks

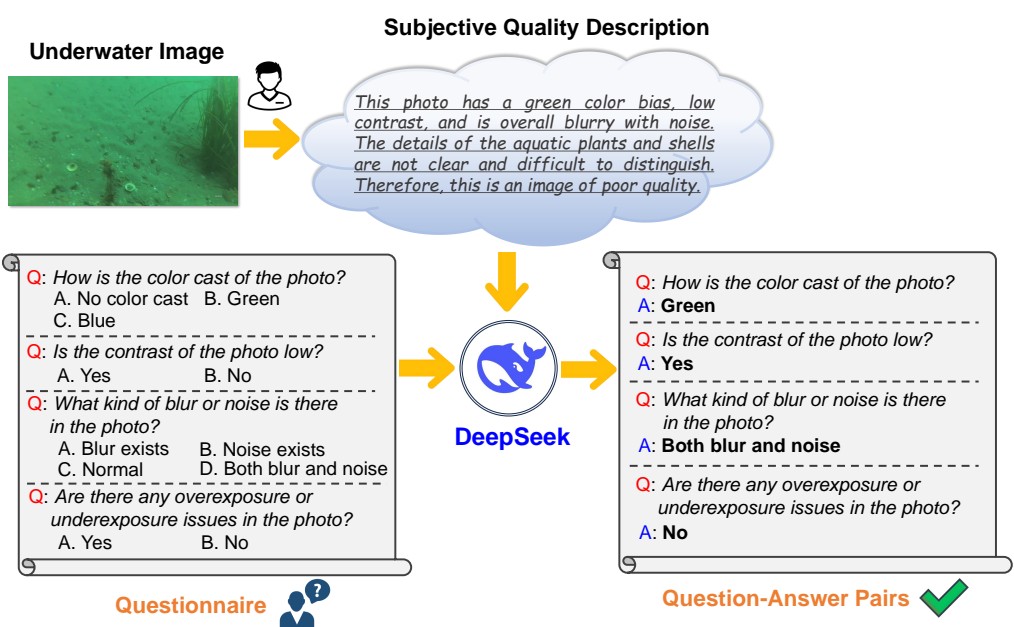

Figure 4: Illustration of generating question-answer pairs of the essential attributes according to the subjective description by DeepSeek.

of advanced vision, such as image description Agrawal et al. (2019); Chen et al. (2015); Young et al. (2014), visual question and answering Antol et al. (2015); Marino et al. (2019); Schwenk et al. (2022), and language-related applications Li et al. (2023); Liu et al. (2024c), etc. Nevertheless, MLLMs still exhibit weaknesses in dealing with low-level perceptual issues Wu et al. (2023a), which is worthy of further investigation.

## 3 THE MUIQD-DESCRIPTION DATASET

The MUIQD-Description dataset is specifically constructed to benchmark the quality description ability of MLLMs for underwater images. In the following, we introduce the construction process of MUIQD-Description in detail.

### 3.1 PREPARATION OF UNDERWATER IMAGES

We collected a total of 18634 authentic underwater images from the subsets of existing underwater image databases Lian et al. (2023); Liu et al. (2024e); Li et al. (2019); Islam et al. (2020a;b). The detailed information for sampling images from each dataset is listed in Table 1. Apart from removing a small number of duplicate images, we almost used all images from each image dataset to construct MUIQD. The collected images cover abundant underwater scenes, such as fish, coral, rock, plant, diver and man-made objects, etc. In addition, the images involve typical underwater quality degradations, including color cast, low contrast, insufficient luminance, blur, foggy, etc. The diversities across the image content and quality level ensure that the MUIQD-Description dataset serves as a robust benchmark for testing MLLMs on underwater image quality perception. In Fig. 1, we show the quality distribution of all images measured by the UCIQE metric Yang & Sowmya (2015) with the values normalized to [0,1]. From this figure, we clearly observe that the underwater image quality spans a wide range, demonstrating the favorable quality diversities of the underwater images intuitively.

| Metric | Score | Criterion |
|---|---|---|
| Precision | 0 | The attributes and quality judgements in two descriptions are **completely inconsistent**. |
| | 1 | The attributes and quality judgements in two descriptions are **partially consistent**. |
| | 2 | The attributes and quality judgements in two descriptions are **completely consistent**. |
| Completeness | 0 | The predicted description **does not include any quality information** provided by the reference description. |
| | 1 | The predicted description **includes partial quality information** provided by the reference description. |
| | 2 | The predicted description **includes all quality information** provided by the reference description. |
| Relevance | 0 | The attributes and details in the predicted description are **completely irrelevant** to that of the reference description. |
| | 1 | The attributes and details in the predicted description are **partially relevant** to that of the reference description. |
| | 2 | The attributes and details in the predicted description are **completely relevant** to that of the reference description. |

Table 2: The criterion to calculate the values of Precision, Completeness and Relevance.

## 3.2 SUBJECTIVE EXPERIMENT

After image preparation, we conducted rigorous subjective tests to annotate the underwater image quality. As suggested in Wu et al. (2024), we ask subjects to provide each image with a detailed description (in natural language) of the essential quality attributes (color cast, contrast, blur and noise, exposure, etc.) and details, and a final quality level deduced by the attributes and details. Such quality descriptions record the evaluations of the main factors that affect the image quality and the reason that subjects conclude the image quality level, which are favorable for the MLLMs to comprehend and learn the way that humans perceive the underwater image quality.

We invited 30 subjects to participate in the subjective tests, which are all college students aged from 18 to 25. Their majors are computer science or electronic information and more than 60% of them have prior experience in image quality assessment. The subjective experiments were carried out in a strictly-controlled laboratory environment. The underwater images were evenly distributed to these 30 subjects for annotations. Before the formal test, a training procedure was introduced to make the subjects familiar with the process and rules of the experiment. In addition, to avoid inaccurate descriptions caused by visual fatigue, each observer was suggested to have a 10-minute rest after continuously working for 40 minutes. To improve the efficiency of subjective experiments, we developed an interface to collect the subjects' descriptions, as illustrated in Fig. 2. After subjective experiments, we invited one subject to review all the quality descriptions and revise those improper descriptions, ensuring the reliability of the quality annotations effectively.

## 3.3 STATISTICS OF THE SUBJECTIVE DESCRIPTIONS

We provide a brief statistical analysis of the collected quality descriptions, which is visualized in Figure 3, including word cloud, word frequency and the distribution of the description length (the number of words). Specifically, the word cloud figure shows the words that appear most frequently in all the descriptions and the word frequency figure demonstrates the number of times the word appears in the descriptions. From the word cloud figure, we observe that the words provided by the observers closely relate with the underwater image quality, such as "low contrast", "color bias", "detail", "blurriness", etc. Correspondingly, the words, such as "color", "contrast", "noise", etc. have relatively higher frequencies, as shown in Figure 3(b). These observations clearly reveal the most important factors that affect the underwater image quality. In addition, the number of words in the descriptions mainly concentrates from 30 to 60. The average length of all the descriptions

| Model | Fine-tuning | Precision | Completeness | Relevance | Average |
|---|---|---|---|---|---|
| Qwen2.5VL-3B | no (baseline) | 0.954 | 0.9687 | 1.3173 | 1.08 |
| | LoRA | **1.4907** | **1.3013** | **1.744** | **1.512** |
| | FFT | 1.46 | 1.282 | 1.7367 | 1.4929 |
| Qwen2.5VL-7B | no (baseline) | 0.854 | 0.7927 | 1.2787 | 0.9751 |
| | LoRA | **1.5113** | **1.3027** | **1.758** | **1.524** |
| | FFT | 1.448 | 1.2873 | 1.7453 | 1.4935 |
| Ovis2-16B | no (baseline) | 1.1113 | 1.072 | 1.5327 | 1.2387 |
| | LoRA | **1.4393** | 1.2847 | **1.7193** | **1.4811** |
| | FFT | 1.424 | **1.2993** | 1.7067 | 1.4767 |
| GLM-4V-9B | no (baseline) | 0.6427 | 0.7373 | 1.064 | 0.8147 |
| | LoRA | 1.4753 | 1.25 | 1.7407 | 1.4887 |
| | FFT | **1.5147** | **1.3593** | **1.7633** | **1.5458** |
| InternVL2.5-8B | no (baseline) | 0.5267 | 0.6846 | 1.0173 | 0.7429 |
| | LoRA | 1.4487 | 1.2573 | 1.7367 | 1.4809 |
| | FFT | **1.5178** | **1.3547** | **1.7593** | **1.5439** |
| llava-v1.6-vicuna-13B | no (baseline) | 0.5293 | 0.6107 | 0.9593 | 0.6998 |
| | LoRA | 1.3547 | 1.198 | 1.6573 | 1.4033 |
| | FFT | **1.5173** | **1.34** | **1.7567** | **1.538** |
| Gemma3-12B | no (baseline) | 1.064 | 1.008 | 1.5107 | 1.1942 |
| | LoRA | **1.5707** | **1.3967** | **1.7893** | **1.5856** |
| | FFT | 1.518 | 1.3553 | 1.766 | 1.5464 |

Table 3: Quality description performance comparison of different MLLMs and fine-tuning strategies on the test set. The higher values of Precision, Completeness and Relevance indicate better performance. The best performance values of each MLLM are highlighted with boldface.

is 43.4 words, which is almost four times as long as common high-level image captions Wu et al. (2024) and describe the underwater image quality suitably.

# 4 THE MUIQD-VQA DATASET

Multimodal interaction is the core function of MLLMs. To fundamentally examine and enhance the interactive quality perception ability of MLLMs, we further construct the MUIQD-VQA dataset. Since the MUIQD-Description dataset contains detailed quality descriptions, it can be used to generate the question-answer pairs directly. Specifically, we generated five question-answer pairs for each image. The first question for each image is fixed, which is the fundamental question involving quality perception, referring to "*Please describe the quality of this underwater image in detail and provide a conclusion based on your description.*". The corresponding answer was set to the subjective description of this image. The other four questions involve the four essential quality attributes, i.e., color cast, contrast, blur and noise, and exposure. The question types are randomly selected from three types, i.e., Yes-or-No, What and How. For each question, we predefined two answers for the Yes-or-No question and three or four answers for the What and How questions. For each question, we employ DeepSeek to choose the right answer according to the subjective quality descriptions by leveraging its powerful natural language processing capabilities. We visualize this process with Fig. 4. For one image and its four questions corresponding to the four essential attributes, we input the questions, the predefined answers and the corresponding subjective description into DeepSeek to determine the right answers, generating the final question-answer pairs. Through this manner, we obtained a total of 74536 ($18634 \times 4$) question-answer pairs. We also invited subjects to examine the answers given by the DeepSeek for maintaining reliability. Along with the first question-answer pair, we got 93170 ($18634 \times 5$) question-answer pairs, which were used to examine and calibrate the MLLMs for handling different quality queries.

| Model | Fine-tuning | Yes-or-No | What | How | Average |
|---|---|---|---|---|---|
| Qwen2.5VL-3B | no (baseline) | 65.98% | 39.43% | 38.84% | 48.08% |
| | LoRA | 74.40%+8.42% | **72.68%**+33.25% | 70.42%+31.58% | 72.50%+24.42% |
| | FFT | **78.31%**+12.33% | 72.48%+33.05% | **72.87%**+34.03% | **74.55%**+26.47% |
| Qwen2.5VL-7B | no (baseline) | 67.74% | 69.01% | 56.36% | 64.37% |
| | LoRA | 75.65%+7.91% | **81.46%**+12.45% | **78.03%**+21.67% | 78.38%+14.01% |
| | FFT | **80.06%**+12.32% | 80.01%+11.00% | 76.23%+19.87% | **78.77%**+14.40% |
| Ovis2-16B | no (baseline) | 64.23% | 70.29% | 53.35% | 62.69% |
| | LoRA | **82.31%**+18.08% | **83.70%**+13.41% | **79.18%**+25.83% | **81.73%**+19.04% |
| | FFT | 71.96%+7.73% | 82.65%+12.36% | 79.03%+25.68% | 77.88%+15.19% |
| GLM-4V-9B | no (baseline) | 64.63% | 68.99% | 62.41% | 65.34% |
| | LoRA | **81.91%**+17.28% | 73.48%+4.49% | 72.87%+10.46% | 76.09%+10.75% |
| | FFT | 79.71%+15.08% | **77.42%**+8.43% | **75.98%**+13.57% | **77.70%**+12.36% |
| InternVL2.5-8B | no (baseline) | 53.56% | 32.40% | 27.98% | 37.97% |
| | LoRA | 77.30%+23.74% | 75.87%+43.47% | 68.97%+40.99% | 74.05%+36.08% |
| | FFT | **84.07%**+30.51% | **84.95%**+52.55% | **80.68%**+52.70% | **83.23%**+45.26% |
| llava-v1.6-vicuna-13B | no (baseline) | 60.97% | 50.65% | 28.43% | 46.68% |
| | LoRA | 80.16%+19.19% | 80.91%+30.26% | 74.92%+46.49% | 78.66%+31.98% |
| | FFT | **82.21%**+21.24% | **84.10%**+33.45% | **80.28%**+51.85% | **82.20%**+35.52% |
| Gemma3-12B | no (baseline) | 74.40% | 70.84% | 65.32% | 70.19% |
| | LoRA | 69.69%-4.71% | 58.82%-12.02% | 56.86%-8.46% | 61.79%-8.40% |
| | FFT | **82.97%**+8.57% | **77.82%**+6.98% | **76.88%**+11.56% | **79.22%**+9.03% |

Table 4: Accuracy rate comparison of answering questions about the image essential attributes on the test set. The best performance values of each MLLM are highlighted with boldface. The increments of the accuracy rate against the baseline are marked in red.

## 5 FINE-TUNING OF MLLMS

Fine-tuning of large models is the critical step to adapt them to specific tasks or domains. With the increasing scale of models, fine-tuning techniques for large models have evolved from the initial full fine-tuning (FFT) Devlin et al. (2019) manner which adjusts all parameters to the Parameter-Efficient Fine-Tuning (PEFT) manners, including Adapter Tuning Houlsby et al. (2019), Prompt Tuning Lester et al. (2021), Prefix-Tuning Li & Liang (2021), LoRA Hu et al. (2022), which adjust partial parameters while maintaining the model's generalization capability. Without loss of generality, in this work, we consider both the FFT and PEFT strategies to adapt MLLMs to the underwater image quality perception task. LoRA was selected as the representative PEFT method.

## 6 EXPERIMENTAL RESULTS

### 6.1 EXPERIMENTAL PROTOCOL

In this section, we fully examine the ability of MLLMs for underwater image quality perception. Two kinds of fundamental abilities are considered. The first one refers to the description ability of the underwater image quality, which is to examine whether the MLLMs can describe the quality of a given underwater image accurately. The other one we examine the accuracy of answering questions about the essential attributes, i.e., color cast, contrast, blur and noise, and exposure, which examines the ability of MLLMs for perceiving the essential quality attributes. Seven SOTA MLLMs were employed in this test, which are Qwen2.5VL-3B Bai et al. (2025), Qwen2.5VL-7B Bai et al. (2025), Ovis2-16B Lu et al. (2024), glm-4v-9B Hong et al. (2024), InternVL2.5-8B Chen et al. (2024), llava-v1.6-vicuna-13B Liu et al. (2024b) and gemma3-12B Team et al. (2025). Note that all MLLMs were downloaded from https://huggingface.co/.

For the first examination, we employ three metrics, i.e., Precision, Completeness and Relevance, to measure the similarity between two descriptions in terms of natural language, as suggested in Wu et al. (2023a). To compute these three measures, we input the predicted description and the ground-

| Fine-tuning | Training set | Yes-or-No | What | How | Average |
|---|---|---|---|---|---|
| no (baseline) | None | 67.74% | 69.01% | 56.36% | 64.37% |
| LoRA | Description qa set | 75.25% | 56.48% | 64.06% | 65.26% |
| | Whole set | 75.65% | 81.46% | 78.03% | 78.38% |
| FFT | Description qa set | 66.28% | 68.00% | 64.91% | 66.40% |
| | Whole set | 80.06% | 80.01% | 76.23% | 78.77% |

Table 5: Accuracy rate comparison of no fine-tuning, fine-tuning with the description question-answer (qa) set and the whole training set.

| Fine-tuning | Training set | Precision | Completeness | Relevance | Average |
|---|---|---|---|---|---|
| no (baseline) | None | 0.854 | 0.7927 | 1.2787 | 0.9751 |
| LoRA | Attribute qa set | 0.906 | 0.874 | 1.318 | 1.0327 |
| | Whole set | 1.5113 | 1.3027 | 1.758 | 1.524 |
| FFT | Attribute qa set | 1.3973 | 1.1327 | 1.66 | 1.3967 |
| | Whole set | 1.448 | 1.2873 | 1.7453 | 1.4935 |

Table 6: Quality description performance comparison of no fine-tuning, fine-tuning with the attribute question-answer (qa) set and the whole training set.

truth description into DeepSeek with predefined criterion, as defined in Table 2. For the second test, we compute the accuracy rate of answering the questions of the essential attributes directly. Since MUIQD-Description is integrated into MUIQD-VQA (the first question and answer), we only used MUIQD-VQA to conduct this test. Specifically, we randomly partitioned MUIQD-VQA into a training set, a validation set, and a test set, where the validation set and the test set consist of 1500 images, respectively. The remaining images make up the training set. These three datasets are all non-overlapped.

## 6.2 IMPLEMENTATION CONFIGURATIONS

We fine-tuned MLLMs using the swift-3.4.0 framework Zhao et al. (2025) on an Ubuntu 22.04.5 LTS system with a GPU configured with four A800 80G. During training, we froze the visual module of the MLLM and only trained its language module to ensure that its general feature extraction capabilities were well retained. Some representative parameters were configured as follows. We set the learning rate to 1e-5, per-device-train-batch-size to 2, gradient-accumulation-steps to 2, lora-rank to 8, and lora-alpha to 16, respectively.

## 6.3 PERFORMANCE COMPARISON ACROSS DIFFERENT MLLMS

After fine-tuning, we test the MLLMs on the test set. The experimental results are listed in Table 3 and Table 4. For comparison, we also include the baseline MLLMs without fine-tuning. By observing Table 3, we find that most of the performance values of all the baseline MLLMs are much higher than 0, which indicates that the baseline MLLMs already have a certain level of quality perception ability due to the comprehensive pretraining on a large corpus of data. By fine-tuning on the proposed MUIQD-VQA dataset, no matter LoRA or FFT, we observe that the performance of all the MLLMs achieves significant improvements consistently, which strongly verifies that the proposed MUIQD-VQA dataset can boost the ability of MLLMs for underwater image quality perception.

In Table 4, we report the accuracy rate of answering questions about the images' essential properties, classified by each question type. The increments of the accuracy rate against the baseline are marked in red. From Table 4, similar observations can be obtained that the baseline MLLMs have certain abilities to answer these attribute-related queries. In addition, most of the accuracy rate after fine-tuning is notably higher than that of the baseline, which demonstrates that the perception abilities for the quality attributes of the MLLMs after fine-tuning are highly perfected. It's also observed that the accuracy rate of Gemma3-12B after loRA fine-tuning is lower than that of its baseline model.

This can be attributed that the baseline Gemma3-12B already has superior perception ability for the image attributes as it achieves the highest accuracy rate among all the baseline MLLMs. For such a strong baseline, LoRA that introduces an extra and low-rank matrix for fine-tuning maybe struggle to adapt the MLLM for new tasks and even destroy its original knowledge structure from pretraining, leading to performance degradations. These observations fully verify that the MLLMs can be adapted to underwater image quality perception successfully.

### 6.4 ABLATION STUDIES

To gain a deeper understanding of the MLLMs in quality perception, we performed some necessary ablation studies. Qwen2.5VL-7B was selected as the experimental subject. Here, we are interested in examining the ability of the MLLM for question answering of essential attributes by fine-tuning it with the quality descriptions and vise versa. Toward this end, we fine-tuned MLLM on the first question-answer pairs (description qa set) of the training set and tested it on the other four questions of the test set. In reverse, we fine-tuned the MLLM on the other four question-answer pairs (attribute qa set) of the training set and tested it on the first question of the test set. The experimental results are reported in Table 5 and Table 6. For comparison, we also include the performance of the baseline model and fine-tuned model with the whole training set. From Table 5, we observe that fine-tuning with the description qa set, no matter LoRA or FFT, can improve the ability of MLLMs to answer the attribute question compared with the baseline model. By observing Table 6, we find that fine-tuning with the attribute qa set can also improve the MLLM for describing the underwater image quality. These observations reveal that quality description and perception for the essential attributes can play mutually reinforcing roles in the underwater image quality perception.

## 7 CONCLUSION

In this paper, we have investigated the issue of underwater image quality perception by the MLLMs. We constructed the first large-scale MUIQD dataset composed of MUIQD-Description and MUIQD-VQA to benchmark and facilitate MLLMs for underwater image quality perception. MUIQD-Description includes 18634 authentic underwater images covering abundant underwater scenes and typical quality degradations. Rigorous subjective experiments were conducted to annotate the images with detailed descriptions about the quality attributes and the overall image quality. For supporting MLLMs to handle different quality queries, we further constructed the first instruction dataset MUIQD-VQA by converting the quality descriptions into more than 93K question-answer pairs with the aid of DeepSeek. Experimental results demonstrate that the proposed MUIQD dataset notably improves the ability of MLLMs for underwater image quality perception. This work fully verifies that MLLMs have great potential to be adapted to underwater image quality perception, which also provides strong evidence for further applications of MLLMs in underwater scenarios. In our future work, we intend to further expand the MUIQD dataset and adopt more effective manners to ensure the dataset quality, based on which multidimensional capabilities of MLLMs for underwater image quality perception will be explored.

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

## A  APPENDIX

### ETHICS STATEMENT

This work adheres to the ICLR Code of Ethics. In this study, no human subjects or animal experimentation was involved. All datasets used, including MUIQD, were sourced in compliance with relevant usage guidelines, ensuring no violation of privacy. We have taken care to avoid any biases or discriminatory outcomes in our research process. No personally identifiable information was used, and no experiments were conducted that could raise privacy or security concerns. We are committed to maintaining transparency and integrity throughout the research process.

### REPRODUCIBILITY STATEMENT

We have made every effort to ensure that the results presented in this paper are reproducible. The source code of this paper has been provided in the Supplementary Material. The experimental setup, including training steps, model configurations, and hardware details, is described in detail in the paper. We have also provided a full description of the MUIQD dataset, to assist others in reproducing our experiments.

We believe these measures will enable other researchers to reproduce our work and further advance the field.

### LLM USAGE STATEMENT

Large Language Models (LLMs) were used to aid in the writing and polishing of the manuscript. Specifically, we used an LLM to assist in refining the language, improving readability, and ensuring clarity in various sections of the paper. The model helped with tasks such as sentence rephrasing, grammar checking, and enhancing the overall flow of the text.

It is important to note that the LLM was not involved in the ideation, research methodology, or experimental design. All research concepts, ideas, and analyses were developed and conducted by the authors. The contributions of the LLM were solely focused on improving the linguistic quality of the paper, with no involvement in the scientific content or data analysis.

The authors take full responsibility for the content of the manuscript, including any text generated or polished by the LLM. We have ensured that the LLM-generated text adheres to ethical guidelines and does not contribute to plagiarism or scientific misconduct.

