# Experimental Results

Table 1. Quality description performance comparison of Qwen2.5VL-7B

| Model | Fine-tuning | Precision | Completeness | Relevance | Average |
|---|---|---|---|---|---|
| Qwen2.5VL-7B | LoRA | 1.5113 | 1.3027 | 1.758 | 1.524 |
| (**freeze visual model**) | FFT | 1.448 | 1.2873 | 1.7453 | 1.4935 |
| Qwen2.5VL-7B | LoRA | 1.5139 | 1.3029 | 1.7627 | 1.5265 |
| (**unfreeze visual model**) | FFT | 1.45 | 1.2920 | 1.7412 | 1.4944 |

Table 2. Accuracy rate comparison of answering questions of Qwen2.5VL-7B

| Model | Fine-tuning | Yes-or-No | What | How | Average |
|---|---|---|---|---|---|
| Qwen2.5VL-7B | LoRA | 75.65% | 81.46% | 78.03% | 78.38% |
| (**freeze visual model**) | FFT | 80.06% | 80.01% | 76.23% | 78.77% |
| Qwen2.5VL-7B | LoRA | 76.89% | 82.3% | 78.18% | 79.12% |
| (**unfreeze visual model**) | FFT | 79.83% | 80.29% | 77.39% | 79.17% |

Table 3. Quality description performance comparison of different MLLMs by BLEU1-4 and ROUGE-L, Higher values of BLEU1-4 and ROUGE-L indicate better performance

| Model | Fine-tuning | BLEU-1 | BLEU-2 | BLEU-3 | BLEU-4 | ROUGE-L |
|---|---|---|---|---|---|---|
| Qwen2.5VL-3B | no(baseline) | 0.3343 | 0.1394 | 0.0638 | 0.0336 | 0.2857 |
| | LoRA | **0.5508** | **0.4062** | **0.3185** | **0.2567** | **0.5384** |
| | FFT | 0.5437 | 0.3964 | 0.3060 | 0.2429 | 0.5297 |
| Qwen2.5VL-7B | no(baseline) | 0.2736 | 0.0978 | 0.0383 | 0.0196 | 0.2143 |
| | LoRA | **0.5545** | **0.4164** | 0.3313 | **0.2701** | **0.5536** |
| | FFT | 0.5409 | 0.3964 | **0.3450** | 0.2429 | 0.5302 |
| Ovis2-16B | no(baseline) | 0.2738 | 0.1076 | 0.0429 | 0.0210 | 0.2276 |
| | LoRA | **0.5500** | 0.4008 | **0.3098** | **0.2481** | **0.5306** |
| | FFT | 0.5543 | **0.4013** | 0.3084 | 0.2468 | 0.5255 |
| GLM-4V-9B | no(baseline) | 0.2500 | 0.0860 | 0.0304 | 0.0154 | 0.2059 |
| | LoRA | **0.5773** | **0.4476** | **0.3674** | **0.3113** | **0.5695** |
| | FFT | 0.5598 | 0.4169 | 0.3299 | 0.2696 | 0.5478 |
| InternVL2.5-8B | no(baseline) | 0.2725 | 0.0933 | 0.0369 | 0.0192 | 0.2158 |
| | LoRA | 0.5023 | 0.3589 | 0.2701 | 0.2114 | 0.5058 |
| | FFT | **0.5253** | **0.3621** | **0.2864** | **0.2266** | **0.5149** |
| llava-v1.6-vicuna-13B | no(baseline) | 0.1583 | 0.0672 | 0.0284 | 0.0139 | 0.1730 |
| | LoRA | 0.5124 | 0.3555 | 0.2568 | 0.1941 | 0.4919 |
| | FFT | **0.5588** | **0.4163** | **0.3280** | **0.2678** | **0.5445** |
| Gemma3-12B | no(baseline) | 0.2117 | 0.0776 | 0.0258 | 0.0123 | 0.1792 |
| | LoRA | 0.3765 | 0.1926 | 0.1098 | 0.0663 | 0.3449 |
| | FFT | **0.5594** | **0.4125** | **0.3239** | **0.2631** | **0.5421** |

Table 4. Prediction performance comparison on the UWIQA database

| Model | SRCC | KRCC | PLCC | RMSE |
|---|---|---|---|---|
| BRISQUE | 0.3456 | 0.2562 | 0.3669 | 0.1415 |
| NFERM | 0.3486 | 0.2595 | 0.3925 | 0.1398 |
| NIQE | 0.4347 | 0.3243 | 0.4687 | 0.1343 |
| IL-NIQE | 0.4686 | 0.3476 | 0.4421 | 0.1364 |
| SNP-NIQE | 0.5516 | 0.4199 | 0.5897 | 0.1228 |
| PIQE | 0.2084 | 0.1492 | 0.3224 | 0.1441 |
| NPQI | 0.6078 | 0.4667 | 0.6361 | 0.1173 |
| dipIQ | 0.0869 | 0.0641 | 0.1369 | 0.1506 |
| HyperIQA | 0.6501 | 0.5040 | 0.6799 | 0.1114 |
| UNIQUE | 0.2496 | 0.1835 | 0.2386 | 0.1476 |
| UCIQE | 0.6271 | 0.4863 | 0.6261 | 0.1185 |
| UIQM | 0.5960 | 0.4563 | 0.5928 | 0.1225 |
| CCF | 0.4456 | 0.3344 | 0.4634 | 0.1348 |
| FDUM | 0.6780 | 0.5289 | 0.6462 | 0.1160 |
| UIQI | 0.7423 | 0.5912 | 0.7412 | 0.1020 |
| Twice-Mix | 0.4727 | 0.3501 | 0.4422 | 0.1289 |
| PIGUIQA | 0.7149 | 0.5726 | 0.7476 | 0.1083 |
| EDANet | 0.8104 | 0.6661 | 0.8156 | 0.0860 |
| LIQE | 0.6100 | 0.4740 | 0.6115 | 0.1152 |
| Q_Align | 0.6952 | 0.5488 | 0.7033 | 0.1101 |
| VisualQuality-R1 | 0.7569 | 0.6333 | 0.7285 | 0.1098 |
| Q_Insight | 0.7883 | 0.6511 | 0.7645 | 0.0990 |
| Qwen2.5VL-3B | 0.7742 | 0.6494 | 0.7638 | 0.1099 |
| Qwen2.5VL-7B | 0.8024 | 0.6569 | 0.8032 | 0.0896 |
| Ovis2-16B | 0.8049 | 0.6574 | 0.8041 | 0.0878 |
| GLM-4V-9B | 0.7808 | 0.6505 | 0.7722 | 0.0996 |
| InternVL2.5-8B | 0.7631 | 0.6323 | 0.7546 | 0.1107 |
| llava-v1.6-vicuna-13B | 0.7783 | 0.6507 | 0.7684 | 0.0998 |
| Gemma3-12B | 0.7764 | 0.6501 | 0.7645 | 0.1094 |

Table 5. Quality description performance comparison of VisualQuality-R1 and Q_Insight

| Model | Precision | Completeness | Relevance | Average |
|---|---|---|---|---|
| Q_Insight | 1.302 | 1.2167 | 1.8773 | 1.4653 |
| Visualquality-R1 | 1.428 | 1.2047 | 1.8573 | 1.4967 |

Table 6. Accuracy rate comparison of answering questions of VisualQuality-R1 and Q_Insight

| Model | Yes-or-No | What | How | Average |
|---|---|---|---|---|
| Q_Insight | 0.7093 | 0.6656 | 0.5504 | 0.6418 |
| Visualquality-R1 | 0.6741 | 0.5906 | 0.6158 | 0.6269 |

# Response to Reviewers

We thank all the reviewers for their valuable reviews and kind suggestions.

**Response for Gy26**
**Thanks for your valuable comments and kind suggestions.**

Q1: We fully agree with the suggestions to ensure the reliability of our proposed MUIQD. In subjective experiments of MUIQD construction, before formal test, we trained all the subjects strictly to meet the experimental requirements. After experiments, we invited one subject to review all the annotations and correct some input errors. These measures ensure the reliability of MUIQD effectively. It's really true that more subjects can ensure MUIQD's reliability more effectively. In our future work, we intend to further expand the proposed MUIQD and invite more than three reviewers to review the quality annotations. More stringent standards will be adopted that the quality description will be considered valid only if all reviewers approve it, which will highly ensure the reliability of the dataset.

Q2: Thanks for your comments. In this paper, we follow the quality description configurations of the paper (Q-Instruct: Improving Low-Level Visual Abilities for Multi-Modality Foundation Models, CVPR 2024), which constructs a dataset for benchmarking MLLM's quality perception ability for images captured in the air. The average length of quality descriptions in Q-instruct is 46.4 words and the authors of Q-instruct point out that this length is 4 times as long as common high-level image captions. Therefore, we consider that 43.4 words in this paper can also describe the image quality comprehensively. In Section 3.3 of the revised paper, we have revised "The average length of all the descriptions is 43.4 words, which can describe the underwater image quality comprehensively" to "The average length of all the descriptions is 43.4 words, which is almost four times as long as common high-level image captions Wu et al. (2024) and describe the underwater image quality suitably", which is more accurately. Please refer to Section 3.3 in the revised paper for details.

Q3&Q4: It's a good suggestion. We unfreeze the visual model and examine the corresponding performance. The experimental results are listed in Table 1 and Table 2 in the supplementary file. Qwen2.5VL-7B is used as the experimental subject. As observed in Table 1 and Table 2, unfreezing the visual model can lead to better performance in both quality description and question answering tasks, indicating the quality perception ability of the MLLM is improved. However, we also observe that the performance gain is limited, which can be attributed to that the visual model of the MLLM already has fairly strong feature extraction ability after pretraining. Unfreezing the visual model in fine-tuning is unable to bring about significant performance improvement. Due to page limit constraints, we put the experimental results in the supplementary file, please refer to Table 1 and Table 2 in the supplementary file for details.

Q5: This is a very valuable query. In this paper, we follow the strategy of Q-instruct (Q-Instruct:

Improving Low-Level Visual Abilities for Multi-Modality Foundation Models, CVPR 2024) which employs GPT to both generate ground-truth QA labels and descriptive quality scores and measure the experimental results. In this paper, we employ DeepSeek to assist to complete these tasks. To verify our conclusions, we further employ five standard measures which are commonly used in the field of natural language processing, i.e., BLEU-1, BLEU-2, BLEU-3, BLEU-4 and ROUGE-L, to measure the quality description performance of MLLMs. These five measures quantify the accuracy of generated text (MLLMs' predicted quality descriptions in this paper) against the ground-truth text (subjectively-annotated quality descriptions in this paper). The accuracy rate comparison results in Table 3 in this paper are objective, which have nothing to do with DeepSeek. We list the new experimental results in Table 3 of the supplementary file. The experimental results show that the fine-tuned MLLMs achieves much better performance than the baseline models measured by BLEU-1, BLEU-2, BLEU-3, BLEU-4 and ROUGE-L indexes, which highly aligns with our original conclusions measured by DeepSeek and thus verifies that our findings are convincing. Due to page limit constraints, we put the new experimental results in Table 3 in the supplementary file. Please refer to Table 3 in the supplementary file for more details.

Q6: Thanks for your careful observations. In Table 3, only the performance of Gemma3-12B after LoRA fine-tuning is obviously worse than that of its baseline model. We can also observe that the baseline Gemma3-12B model achieves the highest performance among all the baseline MLLMs, which indicates that the pre-trained Gemma3-12B model already has superior perception ability for the image attributes. For such a strong baseline, LoRA fine-tuning that introduces an extra and low-rank matrix struggles to adapt the MLLM for the new tasks and even destroy its original knowledge structure from pretraining, leading to performance degradations. We have added these discussions in Section 6.3 in the revised paper. Please refer to Section 6.3 in the revised paper for details.

Q7: Thanks for your kind suggestion. As our dataset is annotated with quality descriptions, rather than the mean opinion scores (MOS), the traditional UIQA models such as EDANet and PIGUIQA that predict the image quality score, can't be evaluated on our dataset. Toward this end, we reconstructed all the fine-tuned MLLMs to predict the image quality score by extracting the last-layer features and regressing them to the image quality score. Then we compared them with the traditional UIQA methods on the dedicated UIQA database UWIQA proposed from the paper (A reference-free underwater image quality assessment metric in frequency domain," Signal Process., Image Commun., vol. 94, 2021, Art. no. 116218). The experimental results are listed in Table 4 in the supplementary file. Experimental results demonstrate that our fine-tuned MLLMs can also achieve superior prediction performance over the dedicated UIQA models. Please refer to Table 4 in the supplementary file for details.

**Response for mHaP**
**Thanks for your valuable comments and kind suggestions.**

W1: This is true that this paper lacks methodological innovation. The aim of this paper is to

verify and further facilitate the capabilities of MLLMs in the specific task of underwater image quality perception, instead of methodological investigation. Toward this end, we construct the first large-scale MLLM-oriented underwater image quality dataset MUIQD and introduced MLLMs to the fundamental underwater vision task, i.e., underwater image quality perception. Through comprehensive and rigorous subjective experiments, we provide detailed quality annotations and generate reliable instructions, making it feasible to tune MLLMs for underwater image quality perception. Experimental results demonstrate that the proposed MUIQD dataset promotes the abilities of MLLMs on underwater image quality perception significantly, which strongly verifies the value and the fundamental contributions of our proposed MUIQD dataset. This paper fully verifies that MLLMs have great potential to be adapted to underwater image quality perception, which also provides strong evidence for further applications of MLLMs in underwater scenarios.

W2&Q1: Thank you for your careful review. We didn't provide a detailed sampling introduction in the paper. In fact, we almost used all the images of each involved image database. Specifically, we only removed a small number of duplicate images in each database and used the remaining images to construct MUIQD. The involved underwater image databases are widely-adopted in the underwater image processing field, such as underwater image quality assessment, underwater image enhancement, etc. Therefore, the images in MUIQD were not randomly sampled. We have pointed out this issue about image sampling explicitly in Section 3.1 in the revised manuscript. Additionally, we have added Table 1 to illustrate the image number comparison of the image dataset and sampled images. Please refer to Table 1 and Section 3.1 in the revised paper for details.

W3&Q2: This is a very valuable question. In this paper, we follow the strategy of Q-instruct (Q-Instruct: Improving Low-Level Visual Abilities for Multi-Modality Foundation Models, CVPR 2024) which employs GPT to both generate ground-truth QA labels and evaluate the experimental results. In this paper, we employ DeepSeek to assist to complete these tasks. To verify our conclusions, we further employed five standard measures which are commonly used in the field of natural language processing, i.e., BLEU-1, BLEU-2, BLEU-3, BLEU-4 and ROUGE-L, to measure the quality description performance of MLLMs. These five measures quantify the accuracy of generated text (MLLMs' predicted quality descriptions in this paper) against the ground-truth text (subjectively-annotated quality descriptions in this paper). The accuracy rate comparison results in Table 3 in this paper are objective, which have nothing to do with DeepSeek. We list the new experimental results in Table 3 in the supplementary file. The experimental results show that the fine-tuned MLLMs achieves much better performance than the baseline models measured by BLEU-1, BLEU-2, BLEU-3, BLEU-4 and ROUGE-L indexes, which highly aligns with our original conclusions measured by DeepSeek and thus verifies that our findings are convincing. Due to page limit constraints, we put the new experimental results in Table 3 in the supplementary file. Please refer to Table 3 in the supplementary file for more details.

Q3: In this paper, we considered the quality attributes by in-depth analysis of underwater imaging characteristics. Specifically, during underwater imaging, different kinds of light have

different attenuation rates. The red light has highest attenuation rate. Therefore, underwater images often exhibit severe color clast, bluish, greenish and their combine. In addition, the backward scattering that the light reflected by the water toward the camera before arriving at the object leads to severe blur and contrast degradation. The insufficient luminance under water introduces exposure problem and noise. Therefore, underwater images mainly suffer from color cast, blur, low contrast, noise and underexposure. In this paper, we considered the above quality attributes, which are representative attributes that affect the underwater image quality. These attributes are also considered as the core features in the existing UIQA methods to characterize the underwater image quality, such as UCIQE, UIQM, UIQI, etc., which verifies the representativeness of the considered attributes in this paper.

W4&Q4: Thanks for your kind suggestion. As our dataset is annotated with quality descriptions, rather than the mean opinion scores (MOS), the traditional computational IQA or UIQA models, such as UCIQE and UIQM, that predict the image quality score, can't be evaluated on our dataset. Toward this end, we reconstructed all the fine-tuned MLLMs to predict the image quality score by extracting the last-layer features and regressing them to the image quality score. Then we compared them with the traditional computational IQA and UIQA methods on a dedicated UIQA database UWIQA proposed from the paper (A reference-free underwater image quality assessment metric in frequency domain," Signal Process., Image Commun., vol. 94, 2021, Art. no. 116218). Due to page limit constraints, we put the experimental results in Table 4 in the supplementary file. Experimental results demonstrate that our fine-tuned MLLMs can also achieve superior prediction performance over the dedicated IQA and UIQA models, which verifies the preferable generalization capabilities of the fine-tuned MLLMs. Please refer to Table 4 in the supplementary file for details.

**Response for UWao**
**Thanks for your valuable comments and kind suggestions.**

Q1&Q3: Thanks for your kind suggestion. The advantages of the proposed MUIQD over prior underwater IQA datasets, such as UIQD, UID2021, UWIQA, etc., mainly lies in the following aspects. The proposed MUIQD-description dataset contains a total of 18634 underwater images, which cover abundant underwater scenes and quality-degradation situations. In addition, we annotated the underwater image essential attributes, details and the overall quality level, rather than only a single number that indicates the image quality adopted by existing underwater image quality databases. What's more, we generate more than 93K question-answer instructions for tuning MLLMs to perceive the underwater image quality more effectively. Therefore, the proposed MUIQD is the first large-scale MLLM-oriented underwater image dataset for benchmarking and facilitating MLLMs for underwater image quality perception. We have added these comparisons and the advantages of the proposed MUIQD in Section 1 in the revised paper. Please refer to Section 1 in the revised paper for details.

Q2: Thanks for your kind suggestions. We have conducted experiments to include advanced IQA methods, such as LIQE, Q_Align, VisualQuality-R1 and Q-Insight, for comparison. Specifically, LIQE and Q_Align predict the image quality score. VisualQuality-R1 and Q-Insight predict both the image quality score and quality descriptions. As our proposed MUIQD is only annotated with quality descriptions without the mean opinion scores (MOS), we compare them with a dedicated UIQA database UWIQA and the proposed MUIQD. Due to the page limits, we put the experimental results in Table 4, Table 5 and Table 6 in the supplementary file. Please refer to Table 4, Table 5 and Table 6 in the supplementary file for details. In addition, we have expanded the related work by introducing the advanced VLM-based IQA models, such as LIQE, Q_Align, VisualQuality-R1 and Q-Insight, etc. in Section 2.1.1 in the revised paper. Please refer to Section 2.1.1 in the revised paper for details.

Q4: Thanks for your careful observations. In Table 3, only the performance of Gemma3-12B after LoRA fine-tuning is obviously worse than that of its baseline model. We can also observe that the baseline Gemma3-12B model achieves the highest performance among all the baseline MLLMs, which indicates that the pre-trained Gemma3-12B model already has superior perception ability for the image attributes. For such a strong baseline, LoRA fine-tuning that introduces an extra and low-rank matrix struggles to adapt the MLLM for the new tasks and even destroy its original knowledge structure from pretraining, leading to performance degradations. In addition, we used uniform parameter configurations, such as the learning rate, the rank value in LoRA to tune all the MLLMs, which may be unsuitable for Gemma3-12B in the LoRA fine-tuning. Therefore, we can adjustment parameters and determine a set of suitable parameters for Gemma3-12B in LoRA finetuning. In addition, we can also employ more advanced PEFT approaches to fine-tune Gemma3-12B, such as DoRA, AdaLoRA, etc. We have added discussions of this issue in Section 6.3 in the revised paper. Please refer to Section 6.3 in the revised paper for details.

**Response for 62Xc**
**Thanks for your valuable comments and kind suggestions.**

Q1: Thanks for your comments. The invited subjects are all college students aged from 18 to 25. Their majors are computer science or electronic information and more than 60% of them have prior experience in image quality assessment. We have added these subjects' background information in Section 3.2 in the revised manuscript. Please refer to Section 3.2 in the revised manuscript for details.

Q2: We have added Table 1 to specify how many images were sourced from each image dataset. In addition, we removed a small number of duplicate images in each database and used the remaining images to construct MUIQD. Therefore, MUIQD doesn't include duplicate images. Please refer to Table 1 in the revised paper for details.

Q3: Thanks for your kind suggestion. The proposed MUIQD still has some limitations, for

example, although it contains 18634 images, the image number is still not enough; in subjective experiments, we only invited one subject to review the quality descriptions, it may cause some bias. In our future work, we plan to invite more than three subjects to review the quality descriptions together, which ensures the reliability of quality annotations more effectively. We have discussed this issue in Section 7 in the revised paper. Please refer to Section 7 in the revised paper for details. The mentioned distortion types, e.g., motion blur from water current, were all included in MUIQD and the images in MUIQD are not biased toward specific underwater scenes.