# OpenReview forum: "MUIQD: Benchmarking and Facilitating Multimodal LLMs for Underwater Image Quality Perception"
_ICLR.cc/2026/Conference — Submitted to ICLR 2026_

### Official Review · Reviewer_62Xc · 2025-10-20

**Soundness:** 3
**Presentation:** 3
**Contribution:** 3
**Rating:** 8
**Confidence:** 5

**Summary:**

This paper addresses a critical and underexplored gap: the application of Multimodal Large Language Models (MLLMs) to underwater image quality perception, a foundational task for underwater vision applications. The proposed large-scale MUIQD dataset consists of MUIQD-Description and MUIQD-VQA subsets. With the proposed MUIQD, the abilities of existing MLLMs were fully examined. Furthermore, by fine-tuning with MUIQD, the abilities of existing MLLMs on underwater image quality perception were improved significantly, which fully demonstrates the value of the proposed MUIQD dataset. This paper is well-structured and the experimental results are sound.

**Strengths:**

(1) The proposed MUIQD dataset is the first large-scale benchmark tailored for MLLMs in underwater image quality perception tasks, consisting of two complementary subsets (MUIQD-Description and MUIQD-VQA). MUIQD-Description’s 18,634 authentic underwater images and detailed subjective annotations well address the major limitation of existing underwater image quality datasets, which only offer single mean opinion scores (MOS). MUIQD-VQA’s 93K+ question-answer pairs further enables evaluation of MLLMs’ interactive quality perception. The proposed MUIQD provides a high-quality platform for benchmarking MLLMs in underwater image quality perception.
(2) This work conducted extensive experiments to validate the proposed dataset’s utility, including: (1) fine-tuning seven SOTA MLLMs with both FFT and LoRA; (2) evaluating performance with task-relevant metrics (Precision/Completeness/Relevance for description, accuracy for attribute-based QA); and (3) ablation studies to reveal mutual reinforcement between quality description and attribute perception. Experimental results consistently demonstrate that the proposed MUIQD significantly boosts MLLMs’ performance, confirming the value of MUIQD and MLLMs’ adaptability to underwater image quality perception tasks.
(3) The paper is well-structured, with clear presentation of the problems, the proposed dataset and the experimental results and analysis.

**Weaknesses:**

(1) The authors mention 30 subjects participated in annotations, but key details about the subjects’ background are missing (e.g., whether they had prior experience in image quality assessment).
(2) The manuscript states images were collected from existing datasets (Lian et al. 2023; Liu et al. 2024e; etc.), but it does not specify how many images were sourced from each dataset, nor whether any duplicate images were included.
(3) The authors should explicitly discuss the dataset’s limitations, for example: does MUIQD-VQA cover all critical quality degradations (e.g., motion blur from water current, which is common in real underwater imaging)? are the images in MUIQD biased toward specific underwater scenes (e.g., coastal vs. deep-sea environments)?

**Questions:**

Please see the weaknesses.

---

> ### Author Response · Authors · 2025-11-25
> **response for MUIQD**
>
> Q1: Thanks for your comments. The invited subjects are all college students aged from 18 to 25. Their majors are computer science or electronic information and more than 60% of them
> have prior experience in image quality assessment. We have added these subjects’ background information in Section 3.2 in the revised manuscript. Please refer to Section 3.2 in the revised manuscript for details.
>
> Q2: We have added Table 1 to specify how many images were sourced from each image dataset. In addition, we removed a small number of duplicate images in each database and used the remaining images to construct MUIQD. Therefore, MUIQD doesn’t include duplicate images. Please refer to Table 1 in the revised paper for details.
>
> Q3: Thanks for your kind suggestion. The proposed MUIQD still has some limitations, for example, although it contains 18634 images, the image number is still not enough; in subjective experiments, we only invited one subject to review the quality descriptions, it may cause some bias. In our future work, we plan to invite more than three subjects to review the quality descriptions together, which ensures the reliability of quality annotations more effectively. We have discussed this issue in Section 7 in the revised paper. Please refer to Section 7 in the revised paper for details. The mentioned distortion types, e.g., motion blur from water current, were all included in MUIQD and the images in MUIQD are not biased toward specific underwater scenes.

---

### Official Review · Reviewer_UWao · 2025-10-27

**Soundness:** 2
**Presentation:** 2
**Contribution:** 2
**Rating:** 4
**Confidence:** 5

**Summary:**

The paper introduces MUIQD, a benchmark for underwater image-quality perception that pairs free-form quality descriptions with attribute-focused VQA. It evaluates seven modern MLLMs using language-based Precision/Completeness/Relevance for descriptions and QA accuracy for attributes, and shows that fine-tuning (LoRA or full-finetune) on MUIQD consistently boosts both abilities, indicating strong transfer between description and attribute QA.

**Strengths:**

1. Separates quality description and attribute QA to probe complementary aspects of perception (content + essential degradations).
2. Tests 7 SOTA MLLMs with clear language-based metrics plus attribute-accuracy.
3. Both LoRA and full fine-tuning on MUIQD yield substantial improvements; cross-task results suggest complementary benefits between description and attribute QA.

**Weaknesses:**

1. Provide a thorough comparison between the proposed dataset and existing underwater image-quality perception datasets, including both quantitative statistics and qualitative examples.
2. Include stronger baselines by adding advanced IQA methods (e.g., VisualQuality-R1) to the comparisons, and expand the Related Work section to cover representative IQA approaches.
3. Clearly articulate the advantages of the proposed dataset over prior underwater IQA datasets, e.g., scale, image and scene diversity, acquisition conditions, annotation protocol and reliability, label granularity, and task relevance, supported by concrete metrics.
4. Explain in detail why fine-tuned models (e.g., Gemma3-12B) underperform their corresponding baselines. Analyze potential causes (data shift, overfitting, optimization choices, instruction/format mismatch, evaluation setup) and discuss diagnostic experiments and remedies.

**Questions:**

See weaknesses.

---

> ### Author Response · Authors · 2025-11-25
> **response for MUIQD**
>
> Q1&Q3: Thanks for your kind suggestion. The advantages of the proposed MUIQD over prior underwater IQA datasets, such as UIQD, UID2021, UWIQA, etc., mainly lies in the following aspects. The proposed MUIQD-description dataset contains a total of 18634 underwater images, which cover abundant underwater scenes and quality-degradation situations. In addition, we annotated the underwater image essential attributes, details and the overall quality level, rather than only a single number that indicates the image quality adopted by existing underwater image quality databases. What’s more, we generate more than 93K question-answer instructions for tuning MLLMs to perceive the underwater image quality more effectively. Therefore, the proposed MUIQD is the first large-scale MLLM-oriented underwater image dataset for benchmarking and facilitating MLLMs for underwater image quality perception. We have added these comparisons and the advantages of the proposed MUIQD in Section 1 in the revised paper. Please refer to Section 1 in the revised paper for details.
>
> Q2: Thanks for your kind suggestions. We have conducted experiments to include advanced IQA methods, such as LIQE, Q_Align, VisualQuality-R1 and Q-Insight, for comparison. Specifically, LIQE and Q_Align predict the image quality score. VisualQuality-R1 and Q-Insight predict both the image quality score and quality descriptions. As our proposed MUIQD is only annotated with quality descriptions without the mean opinion scores (MOS), we compare them with a dedicated UIQA database UWIQA and the proposed MUIQD. Due to the page limits, we put the experimental results in Table 4, Table 5 and Table 6 in the supplementary file. Please refer to Table 4, Table 5 and Table 6 in the supplementary file for details. In addition, we have expanded the related work by introducing the advanced VLM-based IQA models, such as LIQE, Q_Align, VisualQuality-R1 and Q-Insight, etc. in Section 2.1.1 in the revised paper. Please refer to Section 2.1.1 in the revised paper for details.
>
>
> Q4: Thanks for your careful observations. In Table 3, only the performance of Gemma3-12B after LoRA fine-tuning is obviously worse than that of its baseline model. We can also observe that the baseline Gemma3-12B model achieves the highest performance among all the baseline MLLMs, which indicates that the pre-trained Gemma3-12B model already has superior perception ability for the image attributes. For such a strong baseline, LoRA fine-tuning that introduces an extra and low-rank matrix struggles to adapt the MLLM for the new tasks and even destroy its original knowledge structure from pretraining, leading to performance degradations. In addition, we used uniform parameter configurations, such as the learning rate, the rank value in LoRA to tune all the MLLMs, which may be unsuitable for Gemma3-12B in the LoRA fine-tuning. Therefore, we can adjustment parameters and determine a set of suitable parameters for Gemma3-12B in LoRA finetuning. In addition, we can also employ more advanced PEFT approaches to fine-tune Gemma3-12B, such as DoRA, AdaLoRA, etc. We have added discussions of this issue in Section 6.3 in the revised paper. Please refer to Section 6.3 in the revised paper for details.

---

> > ### Comment · Reviewer_UWao · 2025-11-25
> >
> > The detailed rebuttal has addressed my concerns. I will raise my rating.

---

> > > ### Author Response · Authors · 2025-11-25
> > > **response**
> > >
> > > Thanks a lot for your approval of our work and valuable comments, which instruct us to improve our paper significantly.

---

### Official Review · Reviewer_mHaP · 2025-10-28

**Soundness:** 2
**Presentation:** 2
**Contribution:** 2
**Rating:** 4
**Confidence:** 4

**Summary:**

This paper introduces MUIQD, a large-scale dataset designed to benchmark and enhance multimodal large language models for underwater image quality perception. The dataset consists of two subsets: MUIQD-Description, containing 18,634 underwater images annotated with human-written quality descriptions, and MUIQD-VQA, containing over 93K question–answer pairs automatically generated from those descriptions using DeepSeek. The authors fine-tune several SOTA MLLMs with full fine-tuning and LoRA, evaluating them on description similarity and QA accuracy. Results show consistent improvements after fine-tuning, suggesting that MUIQD effectively enhances the quality perception ability of MLLMs for underwater imagery.

**Strengths:**

1. The paper explores a relatively under-examined domain—underwater image quality perception—using MLLMs. Although the methodology is not novel, applying instruction-tuned multimodal learning to this niche, yet practically important, visual domain is uncommon. Establishing a baseline for underwater scenarios broadens the scope of multimodal benchmarking beyond typical terrestrial or aerial imagery domains.
2. In addition, the experimental section is systematic and reasonably reproducible: multiple models (Qwen, Ovis, LLaVA, Gemma, etc.) and fine-tuning schemes (FFT, LoRA) are compared under consistent settings. Results demonstrate consistent performance gains across all tested MLLMs, lending empirical support to the proposed benchmark’s usefulness.

**Weaknesses:**

1. The work combines existing datasets and tools (DeepSeek, LoRA fine-tuning) without methodological innovation.
2. Image selection rules are not described, no sampling or balancing strategy is provided, only “random sampling.”
3. The dual use of DeepSeek for both QA generation and evaluation is problematic and may inflate the observed improvements. No human or cross-model verification is reported.
4. There is no comparison with traditional underwater IQA metrics (e.g., UCIQE, UIQM) to demonstrate generalization.

**Questions:**

1. How were the 18,634 images selected from existing databases? Were they randomly chosen or curated for quality diversity?
2. Why is DeepSeek used as both a QA generator and an evaluator? This creates a risk of evaluator–model alignment and may inflate scores for models that echo DeepSeek’s phrasing or bias.
3. Given the many quality attributes to consider, how do you ensure these sets are representative?
4. Comparing the fine-tuned MLLMs with traditional computational IQA models to better demonstrate the added value of LLMs in underwater image quality perception.

---

> ### Author Response · Authors · 2025-11-25
> **response for MUIQD**
>
> W1: This is true that this paper lacks methodological innovation. The aim of this paper is to verify and further facilitate the capabilities of MLLMs in the specific task of underwater image quality perception, instead of methodological investigation. Toward this end, we construct the first large-scale MLLM-oriented underwater image quality dataset MUIQD and introduced MLLMs to the fundamental underwater vision task, i.e., underwater image quality perception. Through comprehensive and rigorous subjective experiments, we provide detailed quality annotations and generate reliable instructions, making it feasible to tune MLLMs for underwater image quality perception. Experimental results demonstrate that the proposed MUIQD dataset promotes the abilities of MLLMs on underwater image quality perception significantly, which strongly verifies the value and the fundamental contributions of our proposed MUIQD dataset.
>
> W2&Q1: Thank you for your careful review. In fact, we almost used all the images of each involved image database. Specifically, we only removed a small number of duplicate images in each database and used the remaining images to construct MUIQD. The involved underwater image databases are widely-adopted in the underwater image processing field. Therefore, the images in MUIQD were not randomly sampled. We have pointed out this issue about image sampling explicitly in Section 3.1 in the revised paper. Additionally, we have added Table 1 to illustrate the image number comparison of the image dataset and sampled images.
>
> W3&Q2: This is a very valuable question. In this paper, we follow the strategy of Q-instruct (Q-Instruct: Improving Low-Level Visual Abilities for Multi-Modality Foundation Models, CVPR 2024) which employs GPT to both generate ground-truth QA labels and evaluate the experimental results. In this paper, we employ DeepSeek to assist to complete these tasks. To verify our conclusions, we further employed five standard measures which are commonly used in the field of natural language processing, i.e., BLEU-1, BLEU-2, BLEU-3, BLEU-4 and ROUGE-L, to measure the quality description performance of MLLMs. These five measures quantify the accuracy of generated text against the ground-truth text. The accuracy rate comparison results in Table 3 in this paper are objective, which are not computed by DeepSeek. We list the new experimental results in Table 3 in the supplementary file. The experimental results show that the fine-tuned MLLMs achieves much better performance than the baseline models measured by BLEU-1, BLEU-2, BLEU-3, BLEU-4 and ROUGE-L indexes, which highly aligns with our original conclusions measured by DeepSeek and thus verifies that our findings are convincing. Due to page limit constraints, we put the new experimental results in Table 3 in the supplementary file.
>
> Q3: In this paper, we considered the quality attributes by in-depth analysis of underwater imaging characteristics. Specifically, during underwater imaging, different kinds of light have different attenuation rates. The red light has highest attenuation rate. Therefore, underwater images often exhibit severe color clast, i.e., bluish, greenish and their combine. In addition, the backward scattering that the light reflected by the water toward the camera before arriving at the object leads to severe blur and contrast degradation. The insufficient luminance under water introduces exposure problem and noise. Therefore, underwater images mainly suffer from color cast, blur, low contrast, noise and underexposure. In this paper, we considered the above quality attributes, which are representative attributes that affect the underwater image quality. These attributes are also considered as the core features in the existing UIQA methods to characterize the underwater image quality, such as UCIQE, UIQM, UIQI, etc., which verifies the representativeness of the considered attributes in this paper.
>
> W4&Q4: Thanks for your kind suggestion. As our dataset is annotated with quality descriptions, rather than the mean opinion scores (MOS), the traditional computational IQA or UIQA models, such as UCIQE and UIQM, that predict the image quality score, can’t be evaluated on our dataset. Toward this end, we reconstructed all the fine-tuned MLLMs to predict the image quality score by extracting the last-layer features and regressing them to the image quality score. Then we compared them with the traditional computational IQA and UIQA methods on a dedicated UIQA database UWIQA proposed from the paper (A reference-free underwater image quality assessment metric in frequency domain,” Signal Process., Image Commun., vol. 94, 2021, Art. no. 116218). We put the experimental results in Table 4 in the supplementary file. Experimental results demonstrate that our fine-tuned MLLMs can also achieve superior prediction performance over the dedicated IQA and UIQA models, which verifies the preferable generalization capabilities of the fine-tuned MLLMs.

---

### Official Review · Reviewer_Gy26 · 2025-11-03

**Soundness:** 3
**Presentation:** 3
**Contribution:** 3
**Rating:** 4
**Confidence:** 3

**Summary:**

This paper proposes MUIQD, the first large-scale multimodal dataset designed to enhance the underwater image quality perception capabilities of MLLMs. It consists of two components: MUIQD-Description and MUIQD-VQA. The authors used these annotations to fine-tune multiple MLLMs. Experimental results demonstrate that the proposed MUIQD dataset notably improves the ability of MLLMs for underwater image quality perception and supporting that MLLMs can be adapted for underwater image quality perception. This is crucial for marine exploration and image enhancement tasks. However, there are several aspects of the experimental process that lack rigor, and the authors are expected to provide more reasonable explanations and theoretical justifications.

**Strengths:**

1.	The task motivation is clear and has significant application value. Underwater imaging plays a vital role in a variety of underwater tasks, such as resource exploration, marine monitoring, and biological conservation. The application of MLLMs in underwater scenarios is a relatively under-explored yet valuable direction.
2.	The MLLMs involved in the experiments are comprehensive. Fine-tuning and testing were conducted on multiple representative MLLMs (Qwen2.5VL, Ovis, GLM-4V, InternVL, LLaVA, Gemma, etc.). As shown in Table 2 and Table 3, the data comparison is clear.

**Weaknesses:**

1.	In Section 3.2, after the subjective experiments, the authors invited only one subject to review all the quality descriptions, which may introduce single-reviewer bias. Several subjects should be recruited to review the descriptions collaboratively; for example, each subject could be randomly assigned a subset of the data, and the post-review results could be aggregated. A description should be considered valid only if it is approved by all reviewers. Such a procedure would more effectively ensure the reliability of the quality annotations.
2.	In Section 3.3, the authors state, “The average length of all the descriptions is 43.4 words, which can describe the underwater image quality comprehensively.” What is the evidence for this claim? If longer descriptions are assumed to be more comprehensive, the word count should correlate positively with completeness. The authors need to explain why 43.4 words are sufficient for a comprehensive description of underwater image quality.
3.	In Section 6.2, during training, the authors froze the visual module of the MLLM to ensure that its general feature-extraction capabilities were well retained. However, the visual module extracts image features for the language module; would unfreezing it (e.g., fine-tuning only higher layers or adding LoRA to vision layers) improve detail-perception ability? A more detailed discussion is required.
4.	In Section 6.4 (Ablations), the authors should include comparative experiments and results obtained by unfreezing the visual module, to substantiate the claim made in Section 6.2 that “freezing the visual module ensures that its general feature-extraction capabilities were well retained.”
5.	In Section 6.1, the authors use DeepSeek to score the experimental results. Since DeepSeek itself is an AI model, is AI-scoring-AI reliable? The ground-truth QA labels and descriptive quality scores partly originate from the same model family used for fine-tuning and evaluation; consequently, the observed improvements may reflect the model’s ability to imitate or align with DeepSeek’s linguistic style and preferences rather than a genuine enhancement in low-level visual perception.
6.	In Table 3, Gemma3-12B performs worse than its baseline after LoRA fine-tuning across all metrics. The authors should provide an explanation and discussion of this anomalous result.
7.	Experimentally, the authors should conduct aligned comparisons on the same test set with dedicated underwater image quality assessment (UIQA) models such as EDANet (arXiv:1809.06323) and PIGUIQA (arXiv:2412.15527), both of which have open-source implementations.

**Questions:**

Please see the weaknesses above.

---

> ### Author Response · Authors · 2025-11-25
> **response for MUIQD**
>
> Q1: We fully agree with the suggestions. Before formal subjective test, we trained all the subjects strictly to meet the experimental requirements. After experiments, we invited one subject to review all the annotations and correct some input errors. These measures ensure the reliability of MUIQD effectively. It’s really true that more subjects can ensure MUIQD’s reliability more effectively. In our future work, we intend to expand MUIQD and invite more than three reviewers to review the quality annotations. More stringent standards will be adopted, which will highly ensure the reliability of the dataset.
>
> Q2: Thanks for your comments. In this paper, we follow the quality description configurations of the paper (Q-Instruct: Improving Low-Level Visual Abilities for Multi-Modality Foundation Models, CVPR 2024), which constructs a dataset for benchmarking MLLM’s quality perception ability for images captured in the air. The average length of quality descriptions in Q-instruct is 46.4 words and the authors of Q-instruct point out that this length is 4 times as long as common high-level image captions. Therefore, we consider that 43.4 words in this paper can also describe the image quality comprehensively. In Section 3.3 in the revised paper, we have revised corresponding texts.
>
> Q3&Q4: It’s a good suggestion. We unfreeze the visual model and examine the corresponding performance. The experimental results are listed in Table 1 and Table 2 in the supplementary file. Qwen2.5VL-7B is used as the experimental subject. Experimental results show unfreezing the visual model can lead to better performance in both quality description and question answering tasks, indicating the quality perception ability of the MLLM is improved. However, we also observe that the performance gain is limited, which can be attributed to that the visual model of the MLLM already has fairly strong feature extraction ability after pretraining. Unfreezing the visual model in fine-tuning is unable to bring about significant performance improvement. Due to page limit constraints, we put the experimental results in the supplementary file.
>
> Q5: This is a very valuable query. In this paper, we follow the strategy of Q-instruct which employs GPT to both generate ground-truth QA labels and descriptive quality scores and measure the experimental results. In this paper, we employ DeepSeek to assist to complete these tasks. To verify our conclusions, we further employ five standard measures which are commonly used in the field of natural language processing, i.e., BLEU-1, BLEU-2, BLEU-3, BLEU-4 and ROUGE-L, to measure the quality description performance of MLLMs. These five measures quantify the accuracy of generated text against the ground-truth text. The accuracy rate comparison results in Table 3 in this paper are objective, which are not computed by  DeepSeek. We list the new experimental results in Table 3 in the supplementary file. The experimental results show that the fine-tuned MLLMs achieves much better performance than the baseline models measured by BLEU-1, BLEU-2, BLEU-3, BLEU-4 and ROUGE-L indexes, which highly aligns with our original conclusions measured by DeepSeek and thus verifies that our findings are convincing. We put the new experimental results in Table 3 in the supplementary file. Please refer to the supplementary file for more details.
>
> Q6: Thanks for your careful observations. In Table 3, only the performance of Gemma3-12B after LoRA fine-tuning is obviously worse than that of its baseline model. We can also observe that the baseline Gemma3-12B model achieves the highest performance among all the baseline MLLMs, which indicates that the pre-trained Gemma3-12B model already has superior perception ability for the image attributes. For such a strong baseline, LoRA fine-tuning that introduces an extra and low-rank matrix struggles to adapt the MLLM for the new tasks and even destroy its original knowledge structure from pretraining, leading to performance degradations. We have added these discussions in Section 6.3 in the revised paper.
>
> Q7: Thanks for your kind suggestion. As our dataset is annotated with quality descriptions, rather than the mean opinion scores (MOS), the traditional UIQA models such as EDANet and PIGUIQA that predict the image quality score, can’t be evaluated on our dataset. Toward this end, we reconstructed all the fine-tuned MLLMs to predict the image quality score by extracting the last-layer features and regressing them to the image quality score. Then we compared them with the traditional UIQA methods on the dedicated UIQA database UWIQA proposed from the paper (A reference-free underwater image quality assessment metric in frequency domain,” Signal Process., Image Commun.). The experimental results are listed in Table 4 in the supplementary file. Experimental results demonstrate that our fine-tuned MLLMs can also achieve superior prediction performance over the dedicated UIQA models.

---

### Meta-Review · Area_Chair_GSKL · 2026-01-05

**Summary:**

This paper proposes MUIQD, a dataset designed for underwater image quality perception and evaluation with multimodal large language models. The dataset is intended to support both benchmarking and fine tuning of MLLMs for underwater image quality assessment by providing quality descriptions and attribute-based question–answer pairs. The overarching goal is to enable MLLMs to perceive, describe, and reason about underwater image quality, and to establish a domain-specific benchmark for this task.

Among the reviewers, three initially provided negative scores around the rejection threshold, while one reviewer gave a positive score. During the discussion phase, two reviewers did not further respond to the authors’ rebuttal, and one reviewer explicitly indicated an intention to raise their score to a positive level after reading the authors’ responses. As the Area Chair, I carefully read the paper, all reviewer comments, and the authors’ rebuttal.

Several reviewers acknowledged that the topic is practically relevant and that underwater image quality perception is an under-explored yet meaningful application domain for multimodal models. However, multiple substantive concerns were raised regarding the reliability of the annotations, the strength of the experimental evidence, and the overall depth of analysis.

**Reviewer Concerns:**

One particularly important concern, raised by Reviewer Gy26, relates to the annotation protocol. Specifically, after the initial subjective study, only a single reviewer was responsible for reviewing and validating all quality descriptions, which introduces a clear risk of single-reviewer bias. Although the authors acknowledge this limitation and state that involving multiple reviewers will be addressed in future work, this response does not fully resolve the concern for the current version of the dataset. Given that annotation quality is central to the contribution of a dataset paper, this issue remains a significant weakness from both the reviewers’ and the AC’s perspective.

In addition, I agree with Reviewer mHaP’s assessment regarding the limited analytical depth and innovation. While I do not believe that a paper must introduce new methods in order to be publishable, a dataset-centric contribution should provide sufficient analysis to justify its scientific significance. In this work, the paper does not adequately analyze what fundamentally distinguishes underwater image quality perception from general natural image quality perception. It remains unclear whether the observed improvements stem from genuinely domain-specific characteristics of underwater imagery, or whether similar results could be obtained by simply changing the data domain without modifying the modeling or training strategy.

Relatedly, the paper does not sufficiently examine whether underwater image quality perception poses unique challenges beyond differences in visual appearance or description style. For example, the manuscript lacks analysis on whether the training procedure, perceptual attributes, or learning dynamics differ meaningfully from those used in existing image quality datasets for in-air images. Without such analysis, it is difficult to assess whether the problem studied here represents a distinct and significant research challenge, or primarily a dataset substitution within an existing framework.

Overall, while the paper addresses an interesting application domain and presents a substantial dataset, the current version does not fully resolve key concerns regarding annotation reliability, analytical depth, and the scientific significance of the problem formulation. For these reasons, I believe that the paper has not yet reached the level required for acceptance.

**Reviewer Scores:**

Based on the discussion and the authors’ responses, one reviewer explicitly indicated an intention to increase their score to a positive rating. Two reviewers did not engage further during the discussion, and given that their core concerns were only partially addressed, it is unlikely that their scores would have changed substantially. The remaining reviewer already provided a strong positive score and would likely have maintained it.

---

### Decision · Program_Chairs · 2026-01-26

Reject